# Fascin1 empowers YAP mechanotransduction and promotes cholangiocarcinoma development

Arianna Pocaterra [1], Gloria Scattolin [1], Patrizia Romani[1], Cindy Ament[2], Silvia Ribback[3], Xin Chen [4], Matthias Evert[3], Diego F. Calvisi[3] & Sirio Dupont [1✉]

Mechanical forces control cell behavior, including cancer progression. Cells sense forces through actomyosin to activate YAP. However, the regulators of F-actin dynamics playing relevant roles during mechanostransduction in vitro and in vivo remain poorly characterized. Here we identify the Fascin1 F-actin bundling protein as a factor that sustains YAP activation in response to ECM mechanical cues. This is conserved in the mouse liver, where Fascin1 regulates YAP-dependent phenotypes, and in human cholangiocarcinoma cell lines. Moreover, this is relevant for liver tumorigenesis, because Fascin1 is required in the AKT/NICD cholangiocarcinogenesis model and it is sufficient, together with AKT, to induce cholangio-cellular lesions in mice, recapitulating genetic YAP requirements. In support of these findings, Fascin1 expression in human intrahepatic cholangiocarcinomas strongly correlates with poor patient prognosis. We propose that Fascin1 represents a pro-oncogenic mechanism that can be exploited during intrahepatic cholangiocarcinoma development to overcome a mechanical tumor-suppressive environment.

[1] Department of Molecular Medicine, University of Padua Medical School, Padua, Italy. [2] Institute of Pathology, University of Regensburg, Regensburg, Germany. [3] Institute of Pathology, University of Greifswald, Greifswald, Germany. [4] Department of Bioengineering and Therapeutic Sciences and Liver Center, University of California, San Francisco, California, USA. ✉email: sirio.dupont@unipd.it

Tissue mechanical properties are increasingly considered as critical regulators of cell behavior[1,2]. Cells are subjected to multiple forces such as stretching or compression due to tissue deformation, shearing by liquids flowing at the cell surface, and visco-elastic forces by the extracellular matrix (ECM). Cells can sense these forces and actively respond to them by activating intracellular mechanotransduction pathways, which in turn drive the most appropriate biological responses. Pioneering work has indicated how ECM stiffness is sufficient to drive cell behavior, including the choice between proliferation, differentiation, or death[3,4]. Moreover, ECM stiffness is also considered to be an essential tissue property that may contribute to the development and progression of cancer[5,6].

ECM mechanotransduction is based on the ability of cells to actively develop actomyosin tension through integrin-mediated adhesions, regulated by RHO GTPases, ROCK (Rho-associated kinases), and MLCK (myosin light chain kinases)[1,7,8]. If the ECM opposes strong visco-elastic resisting forces such as in the case of a stiff ECM, tension across focal adhesions will rise, which will activate a corresponding biological response, often including proliferation and survival. If the ECM instead opposes weak resistance, this will not enable the development of cell tension, leading to cell quiescence and/or apoptosis induction. One prominent mechanism by which ECM forces regulate cell behavior is the regulation of the YAP (yes-associated protein 1) and TAZ (WW-domain transcription regulator 1) proteins[9–11]. YAP and TAZ are orthologous transcriptional coactivators that shuttle between the cytoplasm and the nucleus, where they bind to transcription factors of the TEAD family and regulate transcription[12]. Among their functions, YAP/TAZ play a prominent role as oncogenic factors in multiple tissues[13]. However, what actin-regulatory factors are important for ECM mechanotransduction and are relevant to drive YAP/TAZ activity in tissues and for cancer development, remains poorly defined.

To date, the in vivo physiological relevance for YAP mechanotransduction is based on experimental alteration of ECM stiffness, for example during tissue fibrosis[14–16], and in the MMP14 metalloprotease knockout mice[17]. However, manipulation of mechanotransduction in these contexts was either indirect or it entailed the regulation of other parallel and possibly confounding parameters. In addition, direct modulation of actin dynamics and cell contractility by genetic means has so far provided phenotypes that are unrelated to YAP/TAZ[18–20], indicating unexpected specificities. CAPZ is a recent and noteworthy exception identified in large-scale screenings as a regulator of Yorkie (the fly YAP/TAZ homolog)[21,22], and as a rheostat of ECM stiffness mechanosensing[23,24]. Inactivation of CAPZ in mouse liver cells enables increased non-muscle myosin activity, leading to YAP activation and its hallmark phenotypes, including organ overgrowth and hepatocyte dedifferentiation into atypical ductal cells[25]. Thus, CAPZ plays a crucial role in ECM mechanotransduction, and its inactivation unveiled the importance of YAP mechanotransduction for liver physiology.

CAPZ is the prototypical F-actin barbed-end capping protein, which restrains actin filament elongation and concurrently stabilizes actin barbed ends[26,27]. During cell migration CAPZ promotes the formation of branched actin networks that form lamellipodia at the migrating front[28–30]. Moreover, CAPZ prevents the formation of filopodia by antagonizing F-actin elongation by Ena/VASP and its stabilization by Fascin1[29,31–33]. Collectively, this body of evidence suggests that CAPZ regulates the balance between two alternative F-actin networks, favoring Arp2/3-dependent branched over bundled F-actin. A similar function of CAPZ has also been observed in yeast, indicating this is not restricted to cell protrusions and cell migration[34].

Here, starting from this "balancing" function of CAPZ, we identify the Fascin1 F-actin bundling protein as a new factor mediating YAP activation in response to ECM mechanical cues. Moreover, we provide evidence that this event represents a pro-oncogenic mechanism favoring intrahepatic cholangiocarcinoma development.

## Results

**Ena/VASP and Fascin1 sustain YAP/TAZ activity.** Evidence for CAPZ regulating the balance between branched and bundled F-actin mainly comes from studies on cell migration, and whether this is relevant for mechanotransduction remains unexplored. We started our investigation by testing the functional relevance of proteins promoting the bundled F-actin formation, such as Formins, Ena/VASP, and Fascin1. We used YAP/TAZ as a downstream read-out of ECM mechanotransduction, and MCF10A mechano-sensitive mammalian epithelial cells since they are an established cell model for investigating the Hippo pathway and for YAP mechanotransduction[10,23,35–37]. The role of Formins downstream of integrin/RHO signaling and in mechanotransduction is well known[1], and inhibition of Formins with the SMIFH2 small molecule in cells cultured on plastic (i.e., on a stiff substratum) resulted in the nuclear exclusion of YAP/TAZ (Fig. 1a) in line with previous data[23]. The role of Ena/VASP proteins is less understood. We transfected cells with F4P-GFP-Mito expressing plasmid, sequestering endogenous Ena/VASP proteins at the mitochondrial surface and thus blocking their function[31]. Expression of F4P-GFP-Mito, but not its A4P-GFP-Mito control, was sufficient to inhibit cell spreading, as shown by phalloidin staining, and caused a consequent inhibition of YAP/TAZ, as gauged by nuclear/cytoplasmic localization (Fig. 1b).

Besides Ena/VASP proteins, another key component regulated by CAPZ is Fascin[29]. Fascin is a highly conserved protein encoded by three orthologous genes in mammals, which promotes in vitro the formation of rigid and contractile bundles[38], and which is found in filopodia and in F-actin bundles around the nucleus[39–43]. *Fascin1* is the isoform with the broadest expression in mice, whereas *Fascin2* and *Fascin3* expression is limited to the retina and testis, respectively[44]. Recent data suggest a role for Fascin1 as a regulator of the Hippo pathway in WM793 melanoma cells and in A549 non-small cell lung cancer cells[45,46]. However, the functional relevance for this regulation has not been addressed neither in vivo nor in the context of mechanotransduction. MCF10A cells express undetectable levels of *Fascin2* and *Fascin3* mRNA, as measured by qPCR (Supplementary Fig. 1a). We therefore knocked-down Fascin1 by RNA interference, which caused the reduction of radial F-actin bundles (Fig. 1c and Supplementary Fig. 1b, c), and a concomitant translocation of YAP/TAZ towards the cytoplasm (Fig. 1D). This was independently confirmed by treating cells with the G2 small-molecule inhibitor of Fascin (Fig. 1e)[47], and in human cholangiocarcinoma cell lines (see Fig. 4 below). Accordingly, inhibition of Fascin1 reduced YAP/TAZ transcriptional activity measured by the established 8XGTIIC-lux luciferase reporter assay in MDA-MB-231 breast cancer cells (Fig. 1f), which display high level of YAP/TAZ activity[10,48], and whose metastatic ability depends on Fascin[47]. Similar results were obtained by monitoring endogenous YAP/TAZ target genes by qPCR in mouse E0771 breast cancer cells stably expressing Fascin1 shRNAs (Fig. 1g), which we used to validate shRNAs to be used in vivo (see Fig. 4 below). Collectively, this data indicates that the pool of bundled F-actin promoted by Ena/VASP and Fascin1 sustains YAP/TAZ activity when cells are on a stiff substratum.

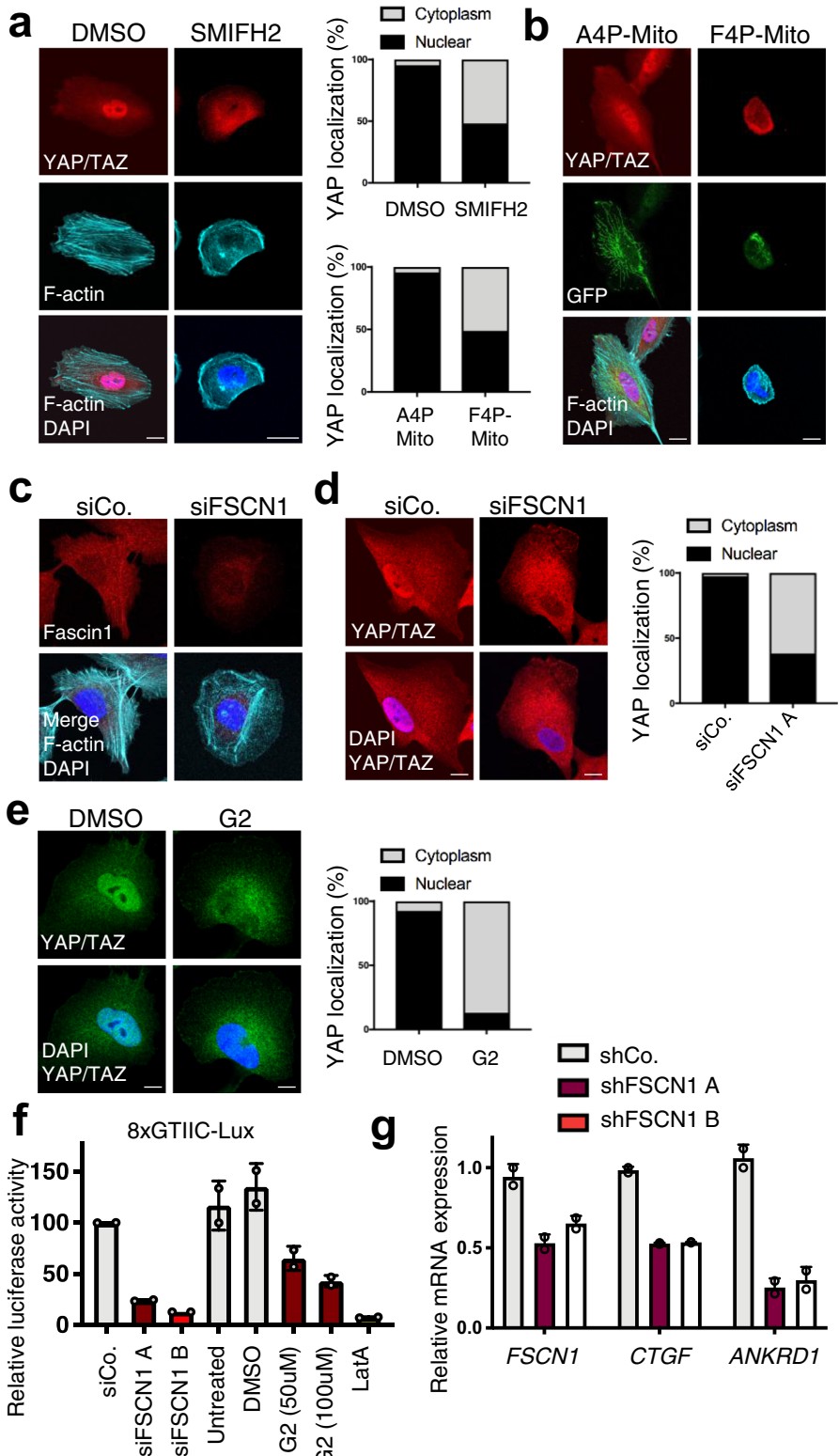

**CAPZ and Arp2/3 antagonize Fascin1-dependent actin during mechanotransduction**. We and others previously described CAPZ as a YAP/TAZ inhibitor in the context of ECM mechanotransduction, but whether this relates with the ability of CAPZ to promote branched F-actin networks and to oppose Fascin1

(see introduction and Supplementary Fig. 2a) remains unknown. To test this hypothesis, we inhibited the Arp2/3 complex, i.e., the master regulator of branched actin, by treating cells plated on a soft ECM—i.e., the condition in which CAPZ is relevant[23]—with the CK-869 small molecule[49]. As shown in Fig. 2a (quantified in

**Fig. 1 Bundled F-actin promoted by Formins, Ena/VASP, and Fascin1 sustains YAP activity on stiff ECM substrata. a** Representative immunofluorescence (IF) of MCF10A cells treated with 60 µM SMIFH2 Formin inhibitor or with the same amount of vehicle (DMSO) for 24 h and stained for YAP/TAZ, F-actin (phalloidin), and DAPI as nuclear counterstain. On the right, quantification of YAP/TAZ subcellular localization, expressed as the percentage of cells in which the nucleus had a stronger staining than the surrounding cytoplasm (Nuclear) or equal/lower than the cytoplasm (Cytoplasm). $n = 3$ (>60 cells per condition in total). $p = 0.0133$ by unpaired Welch's t-test. Scale bar = 10 µm. **b** Representative IF of MCF10A cells transfected with a plasmid encoding for the Ena/VASP inhibitor F4P-GFP-Mito, or for its mutated control A4P-GFP-Mito, and stained for YAP/TAZ, F-actin (phalloidin) and DAPI. On the bottom/left, quantification of YAP/TAZ localization. $n = 3$ (>60 cells per condition in total). $p = 0.0357$ by unpaired Welch's t-test. Scale bar = 10 µm. **c, d** Representative IF of MCF10A cells transfected with FSCN1 siRNA (siFSCN1) or control siRNA (siCo.). In **c**, cells were stained for Fascin1, F-actin (phalloidin), and DAPI. In **d**, cells were stained for YAP/TAZ and DAPI. On the right, quantification of YAP/TAZ localization. Similar results were obtained with siFSCN1 B. $n = 3$ (>60 cells per condition in total). $p = 0.0283$ by unpaired Welch's t-test. Scale bars = 10 µm. **e** Representative IF of MCF10A cells treated with 100 µM G2 Fascin inhibitor or its vehicle (DMSO) for 24 h and stained for YAP and DAPI. On the right, quantification of YAP/TAZ localization. $n = 3$ (>60 cells per condition in total). $p = 0.0088$ by unpaired Welch's t-test. Scale bar = 5 µm. **f** 8XGTIIC-luciferase reporter assay for YAP/TAZ in MDA-MB-231 cells transfected with two independent siRNAs targeting FSCN1 (siFSCN1 A and siFSCN1 B) or control siRNA (siCo.), or treated with different doses of G2 (50 µM or 100 µM), latrunculinA (LatA 0.5 µM) as positive control for inhibition of YAP/TAZ, or DMSO as vehicle. Mean expression in controls was set to 100 and all other samples are relative to this. Data are mean and s.d. **g** qPCR for YAP/TAZ target genes (CTGF and ANKRD1) in mouse EO771 cells stably expressing control short hairpin RNA (shCo.) or two different short-hairpin RNAs targeting FSCN1 (shFSCN1 A and shFSCN1 B). Data are relative to GAPDH expression. Mean expression levels in control cells were set to 1, all other samples are relative to this. Data are mean and s.d.

Fig. 2e), CK-869 treatment rescued YAP/TAZ nuclear localization in cells cultured on a soft ECM, recapitulating the effect of CAPZ depletion (Fig. 2a, quantified in Fig. 2d). CK-869 treatment on plastics, where F-actin bundles are fully empowered by stiffness, did not change YAP/TAZ localization (Supplementary Fig. 2b), similar to previous observations on CAPZ[23]. Overall, this data indicates that CAPZ and Arp2/3, regulators of branched F-actin, contribute to keeping YAP/TAZ out of the nucleus on a soft ECM.

Given the potential competition between branched and bundled F-actin networks[50,51], we then tested whether Arp2/3 and CAPZ modulate YAP/TAZ by regulating Fascin1. In line with this hypothesis, treatment of cells with CK-869 or depletion of CAPZ led to the accumulation of Fascin1-positive structures at cell edges (Fig. 2b and Supplementary Fig. 2b), where CAPZ is usually found (Supplementary Fig. 2c). These structures in part co-localized with Vinculin, a marker for mature focal adhesions (Supplementary Fig. 2d). Moreover, we observed enhanced Fascin1 immunoreactivity in CAPZ-depleted cells plated on a soft ECM (Fig. 2c). Most importantly, Fascin1 activity was required for nuclear YAP/TAZ localization induced by Arp2/3 inhibition or by CAPZ knockdown (Fig. 2d, e). Collectively, these results suggest that on a soft ECM the competition between CAPZ-Arp2/3 and Fascin1 is relevant for the regulation of YAP/TAZ nuclear localization (see scheme in Supplementary Fig. 2a).

To reinforce this idea, overexpressing active S39A Fascin1[52] in cells on a soft ECM promoted partial YAP/TAZ nuclear accumulation (Fig. 2f, see Supplementary Fig. 2e for controls). This did not occur in cells treated with latrunculinA (Supplementary Fig. 2f), indicating an F-actin-dependent function of Fascin1. Finally, we built on the notion that YAP/TAZ are regulated in confluent monolayers by a "mechanical checkpoint" similar to that occurring on a soft ECM, such that inactivation of CAPZ can rescue YAP/TAZ activity in a confluent monolayer[23]. As shown in Supplementary Fig. 2g, overexpressing active Fascin1 was sufficient to rescue YAP/TAZ nuclear localization in confluent cells, and cooperated with oncogenes to relieve contact inhibition of growth. These findings indicate that increased Fascin1 levels and/or activity can promote YAP mechanotransduction in a cell-autonomous manner.

**CAPZ antagonizes Fascin1-dependent YAP activity in the liver.** To challenge the relevance of our observations in vivo, we sought to transpose our observations to the mouse liver, a model system to study YAP/TAZ biology[53,54] and an organ where cells lay in a soft environment[55,56]. Moreover, the hepatocyte-specific

inactivation of Capzb triggers YAP mechanotransduction as well as proliferation and dedifferentiation of hepatocytes into atypical ductal cells (ADC) of cholangiocellular identity[25].

We initially tested whether Fascin1 is downstream of CAPZ also in vivo. For this purpose we inhibited Fascin1 in liver cells by administering the G2 Fascin inhibitor via i.p. injection to CAPZ LKO mice and scored hepatocyte dedifferentiation as a read-out of YAP function[25,54,57]. As shown in Fig. 3a and Supplementary Fig. 3, G2 treatment restricted the expansion of the cholangiocellular marker CK19 in CAPZ LKOs. The effect was partial, likely due to the relatively low affinity of G2 for Fascin1[47]. To test whether Fascin1 was sufficient to activate YAP in vivo, we subsequently overexpressed Fascin1 in the liver of adult wild-type mice using hydrodynamic tail vein (HTV) injection and scored established YAP-induced phenotypes. Expression of Fascin1 was sufficient to increase hepatocyte proliferation, as shown by EdU incorporation (Fig. 3b), and to induce the formation of ADCs, as gauged by staining for the A6 cholangiocellular marker (Fig. 3c). Importantly, these phenotypes were prevented when Yap1/Wwtr1 (TAZ) were knocked out in Fascin1-expressing hepatocytes (Fig. 3c), indicating an effect mediated by activation of YAP/TAZ. These results suggest that CAPZ maintains hepatocyte cell fate by inhibiting Fascin1-dependent YAP activation.

**Fascin1 has a pro-oncogenic function in intrahepatic cholangiocarcinomas.** In the liver, YAP activation not only induces transdifferentiation of hepatocytes into ADCs, but also promotes the development of hepatocellular carcinomas (HCC), cholangiomas and intrahepatic cholangiocarcinomas (iCCA)[58–61], which prompted us to explore the role of Fascin1 in liver tumorigenesis. Overexpression in the mouse liver of myristoylated AKT, an established driver of liver carcinogenesis[62], did not result in any histopathological alteration (Fig. 3d–f). AKT-expressing cells were not detected, implying their elimination, likely due to insufficient fitness in the C57BL/6 N background[63]. In striking contrast, expression of Fascin1 anc AKT was sufficient to induce the appearance of macroscopic nodules on the liver surface (Fig. 3d). The AKT/FSCN1 lesions display enhanced proliferation as gauged by EdU incorporation, they exhibited cholangiocellular features based on the immunoreactivity for the CK19 and A6 markers, and histopathologic analyses indicated them as neoductular proliferation or cholangiomas (Fig. 3e, f). A similar phenotype was observed following expression of AKT and activated TAZ (Fig. 3d–f) or YAP[64]. Furthermore, we detected the presence of hepatocellular adenomas or hepatocellular foci, or of mixed lesions, in AKT/FSCN1 and AKT/TAZ livers (Fig. 3e).

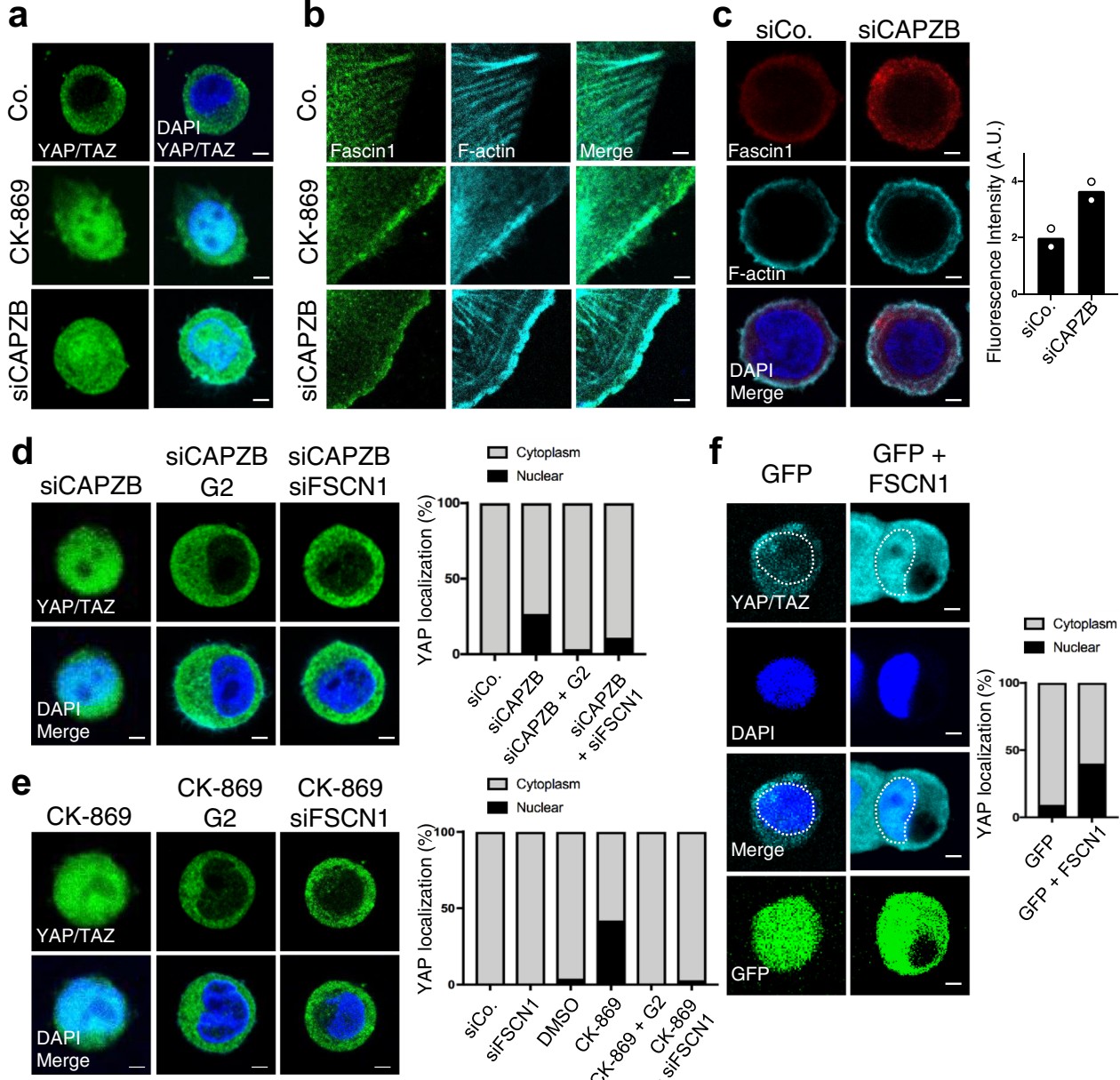

**Fig. 2 CAPZB and Arp2/3 inhibition induce nuclear YAP on soft ECM substrata, which depends on Fascin1. a** Representative IF of MCF10A cells plated on soft Fibronectin-coated polyacrylamide hydrogels ($E = 0.5$ kPa) for one day and then treated for additional 6 h with 80 μM CK-869 Arp2/3 inhibitor or vehicle control (DMSO). Where indicated cells were transfected with CAPZB siRNA (siCAPZB). Cells were stained for YAP/TAZ and DAPI as nuclear counterstain. See quantification of YAP/TAZ localization in **d**. Scale bar = 3 μm. **b** Representative IF of MCF10A cells treated with CK-869 or vehicle control (DMSO) or transfected with CAPZB siRNA (siCAPZB). Cells were stained for Fascin1 and F-actin (phalloidin). Scale bar = 2 μm. **c** Representative IF of MCF10A transfected with CAPZB siRNA (siCAPZB) or control siRNA (siCo.) and replated on soft hydrogels for 48 h. Cells were stained for Fascin1 and F-actin (phalloidin). Quantification of Fascin1 intensity on the right. $n = 2$ independent experiments. Scale bar = 2 μm. **d**, **e** Representative IF of MCF10A cells transfected with CAPZB siRNA (siCAPZB, **d**) or treated with 80 μM CK-869 (**e**) and cultured on soft hydrogels. Where indicated cells were co-treated with the G2 Fascin inhibitor or co-transfected with Fascin1 siRNA (siFSCN1). Cells were stained for YAP/TAZ and DAPI. On the right, quantification of YAP/TAZ localization. $n = 3$ (>60 cells per condition in total). Scale bar = 2 μm. **f** Representative immunofluorescence images of MCF10A cells transfected with GFP or with GFP + FSCN1 plasmids and cultured on soft hydrogels for 24 h. The white dotted line indicates the nucleus. On the right, quantification of YAP/TAZ subcellular localization. $n = 3$ (>60 cells per condition in total). $p = 0.0157$ by unpaired Welch's test. Scale bar = 4 μm.

Finally, we did not observe malignant progression features in the C57BL/6 N background, indicating delayed or reduced progression compared to FVB/N mice[64]. Thus, Fascin1 cooperates with activated AKT to induce liver neoplasia, including cholangiomas.

To explore the functional requirement of Fascin1 in the context of cholangiocarcinoma, we next investigated the levels of Fascin1

in experimental cholangiocarcinomas induced in mice by different oncogene combinations. For this purpose we stained livers of FVB/N mice harboring cholangiocarcinomas induced by HTV injection of AKT together with activated N-Ras-V12D (AKT/N-Ras), with activated Notch Intracellular Domain (AKT/NICD), or with activated YAP-S127A (AKT/YAP)[57,62,64–66]. In

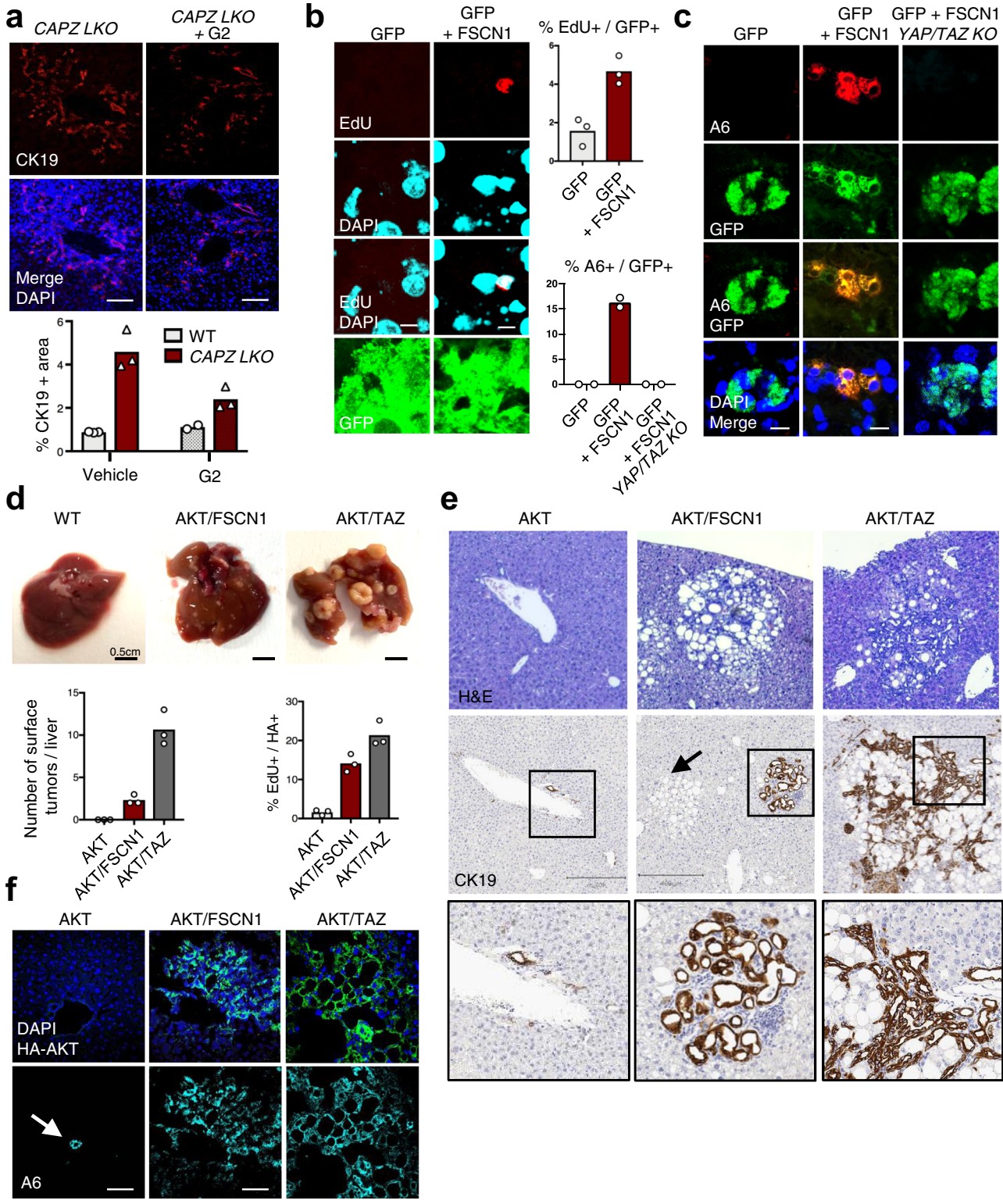

all models tested, a robust Fascin1 immunoreactivity was detected in endothelial cells, and used as an internal control of the staining. However, only the AKT/NICD combination displayed consistent, strong, and diffuse cytoplasmic Fascin1 expression in tumor cells when compared to the surrounding normal tissue (Fig. 4a, b, see higher magnifications in Supplementary Fig. 4a). The absence of Fascin1 overexpression in other oncogenic

combinations likely reflects different underlying molecular mechanisms. Tumors induced by NICD alone were also negative for Fascin1 expression in tumor cells, suggesting the need for combined NICD and AKT signaling (Fig. 4a).

The development of cholangiocarcinomas by AKT/NICD relies on endogenous YAP in cancer cells[57,60], thus representing the ideal experimental set-up to test the requirement for Fascin1. We

**Fig. 3 Fascin1 regulates hepatocyte cell fate through YAP/TAZ and promotes cholangiocarcinoma development. a** Representative immunofluorescence stainings of liver sections from adult tamoxifen-injected *Albumin-CreERT2; Capzbfl/fl* mice (*CAPZ LKO*) mice injected i.p. with the G2 Fascin inhibitor or with vehicle (5% DMSO), stained for the cholangiocellular marker CK19 and DAPI as nuclear counterstain. Analyses were carried out 15 days after tamoxifen injection. On the right, quantification of CK19-positive area in sections of the portal area for the indicated conditions. Mean and single data (mice, n = 3). Scale bar = 100 μm. **b** Representative immunofluorescence stainings of liver sections from mice transduced by hydrodynamic tail vein (HTV) injection with transposon plasmids encoding for GFP or for GFP + Fascin1. Liver sections were stained for EdU. Analyses were carried out 15 days after HTV injection. On the right, quantification of EdU-positive and GFP-positive hepatocytes. Mean and single points (mice, n = 3). Scale bar = 5 μm. **c** Representative immunofluorescence stainings of liver sections from mice transduced by HTV with transposon plasmids encoding for GFP or GFP + Fascin1. YAP/TAZ double KO was obtained by adding the CRE recombinase to the HTV mix, and by injecting *Yap1fl/fl; Wwtr1(TAZ)fl/fl; ROSA26-LSL-lacZ* mice. Analyses were carried out 15 days after HTV injection and induction of CRE activity by tamoxifen. YAP/TAZ recombination was controlled by β-galactosidase staining. Mean and single points (mice, n = 2). Scale bar = 10 μm. **d** Representative pictures and quantification of the macroscopic lesions detected at the liver surface in mice transduced by HTV injection with transposon plasmids expressing myristoylated-HA-AKT alone, together with Fascin1 (AKT/FSCN1), or together with activated TAZ 4SA (AKT/TAZ). Livers were analyzed 7 months after HTV. Mean and single points (mice, n = 3). **e** Representative Hematoxylin and Eosin (H&E) and immunohistochemistry for the cholangiocellular marker CK19 on liver sections from mice transduced by HTV injection as in **d**. The black arrow indicates an example of hepatocellular adenoma. Squares and insets: higher magnification on CK19-positive areas, including cholangiomas and ductular reactions. Livers were analyzed 7 months after HTV. **f** Representative immunofluorescence images of liver sections from mice transduced by HTV injection as in **d**. Liver sections were stained for HA to localize AKT-expressing cells, the cholangiocellular marker A6, and DAPI. The white arrow indicates a normal A6-positive bile duct. Above the pictures: quantification of EdU-positive and HA-positive cells. Cells expressing only AKT were undetectable, likely due to the high resistance to oncogenic transformation of the C57BL/6 N strain. EdU incorporation in this condition refers to non-transduced hepatocytes. Livers were analyzed 7 months after HTV. Average and single points (mice, n = 3). Scale bar = 80 μm.

therefore combined AKT/NICD expression together with control shRNA or with a pre-validated Fascin1 shRNA (see Fig. 1g). Co-expression of AKT/NICD with control shRNA caused the appearance of multiple focal neoplasms as early as 3 weeks after injection (Fig. 4b and Supplementary Fig. 4b), in line with previous evidence[66]. Notably, Fascin1 shRNA caused a significant reduction in the number and size of lesions at this time point (Fig. 4c). This finding indicates the importance of Fascin1 in AKT/NICD-driven cholangiocarcinogenesis. Moreover, Fascin1 functionally cooperates with AKT for the development of cholangiocellular lesions in mice.

Finally, we sought to understand whether these data bear significance for human liver disease. In a panel of human intrahepatic cholangiocarcinoma (iCCA) cell lines, treatment with the G2 Fascin inhibitor caused a dose-dependent inhibition of cell growth (Fig. 4d), recapitulating the effects of YAP/TAZ knockdown (Fig. 4e). Moreover, knockdown of Fascin1 triggered the inhibition of YAP/TAZ nuclear localization (Fig. 4f), indicating a conserved Fascin1-YAP/TAZ axis in human iCCA cells. We then performed Fascin1 immunohistochemistry on a series of human liver cancers with matched normal liver tissues (Supplementary Tables 1 and 2). In the normal human liver, Fascin1 staining was restricted to vascular structures with negative or very low staining in hepatocytes (Fig. 4g), as also seen in mice. We instead observed intense and diffuse cytoplasmic staining in 35/62 (56%) of human iCCA (Fig. 4g), suggesting the overexpression observed in AKT/NICD mouse tumors finds a correlation in the human disease. We also stained human HCC, and in this case only 13/50 (26%) were positive for Fascin1 (Supplementary Fig. 4c). Subsequently, we evaluated a small collection of human iCCA cases with adjoining preinvasive and invasive lesions. Fascin1 immunoreactivity was less frequent in the presinvasive lesions (4/13), where the expression of Fascin1 was predominantly located at the invasive front. Positive Fascin1 staining became more frequent (9/13) and homogeneous in invasive carcinomas (Fig. 4h). All cancer cells were positive for YAP immunostaining suggesting that in human iCCA, like in mouse experimental iCCA (see references above), multiple different mechanisms converge on sustaining YAP levels. Finally, to explore a correlation between Fascin1 expression and prognosis, we estimated the survival curves of patients whose tumor cells stained positive or negative for Fascin1 by the Kaplan–Meier method. Significantly, we found a striking

correlation with poor prognosis in iCCA (Fig. 4i, see also Supplementary Tables 1 and 2), with an estimated median survival of 60 weeks for Fascin1-negative patients and 15 weeks for Fascin1-positive patients. Altogether this data indicates that Fascin1 expression is enhanced in a proportion of human iCCA, in which it might contribute to disease progression and prognosis.

## Discussion

The identification of the YAP mechanotransduction system provided a powerful model to study the effects of tissue mechanical properties on cell behavior. Yet, despite a wealth of knowledge on the most upstream players of this pathway (i.e., at the level of focal adhesions)[1,7,67] and some very recent hints on the mechanisms proximal to YAP regulation[68–71], the relevant intermediate players and the F-actin structures involved remain largely unknown. Moreover, the identity of the pivotal inducers of YAP mechanotransduction in vivo, and their eventual role in cancer progression, remain even less charted territories.

Here, led by the known function of CAPZ in shifting the balance between actin bundled vs. branched structures during cell migration[29,31], we identify Ena/VASP and Fascin1 proteins as required for YAP activity in response to ECM stiffness. On a soft ECM CAPZ and Arp2/3 complexes, promoters of branched F-actin, inhibit YAP activity by limiting the formation of Fascin1-dependent actin. Indeed expression of activated Fascin1 is sufficient to drive nuclear YAP on a soft ECM and in confluent monolayers. The concept of competition between different actin networks in cells has been previously proposed in yeast (where Profilin regulate the balance between Arp2/3 and Formin activity)[51,72], in epithelial cells of liver origin (where actomyosin bundles prevent the formation of Arp2/3 dependent sub-membranous actin)[50], and in axon growth cones (where Arp2/3 activity can restrict myosin-mediated contractility)[73]. Our data suggests that such competition also occurs during ECM mechanotransduction, and becomes relevant when cells are in conditions of decreased ECM stiffness. In these conditions, not only is RHO signaling reduced, leading to reduced activity of ROCK/MLCK and of Formins[1], but at the same time actin structures induced by CAPZ and Arp2/3 outbalance actin bundles promoted by Ena/VASP, Fascin1 and potentially other F-actin bundling proteins. We speculate this might be particularly relevant at intermediate stiffness levels, ultimately tipping the balance in favor or against YAP/TAZ activity.

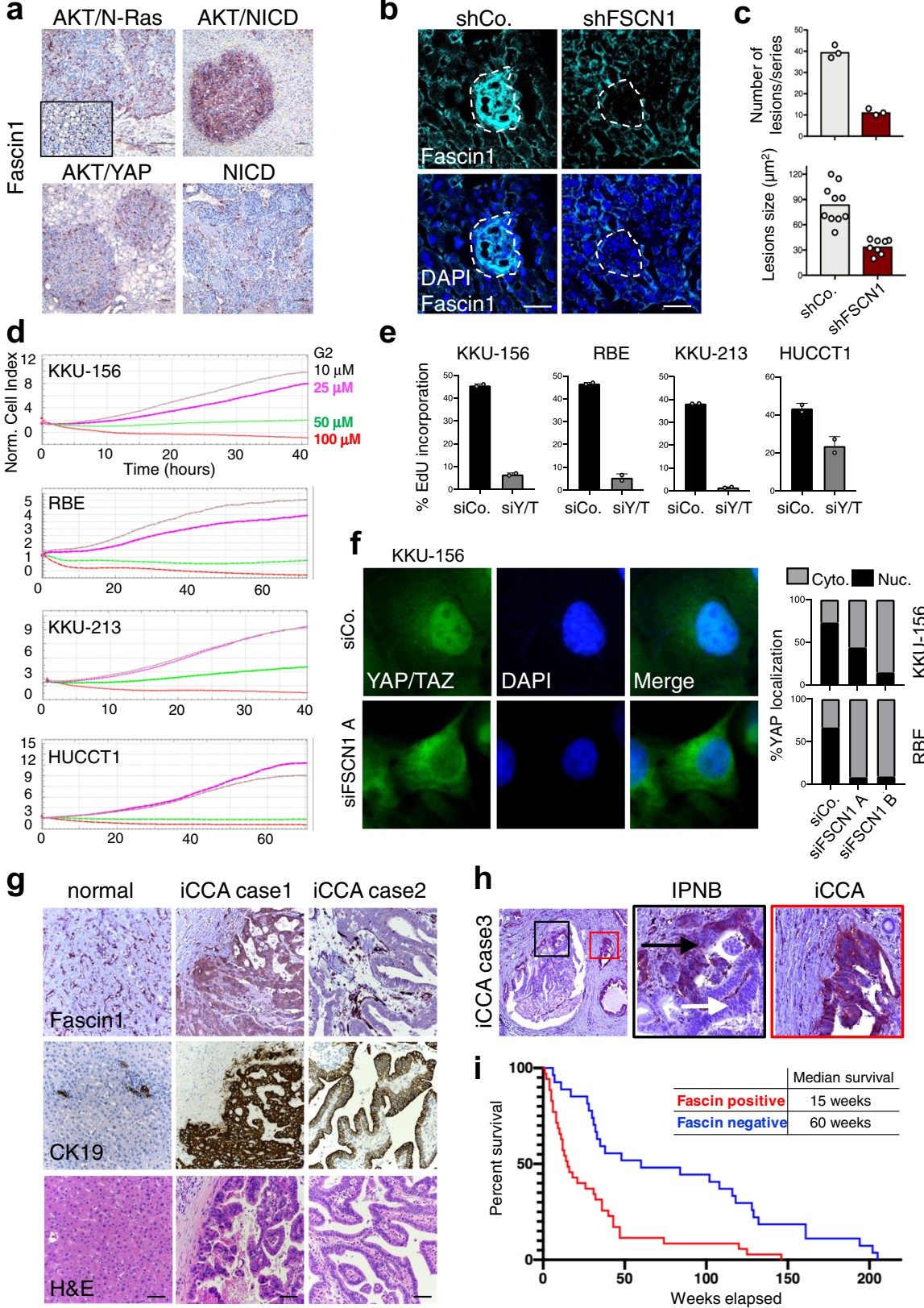

The role of Ena/VASP and Fascin1, key factors driving filopodia formation[39,43], might indicate a specific function of filopodia in the phenotypes we observed. Filopodia are mechanosensitive structures, they support the formation of nascent adhesions that can subsequently develop into mature focal adhesions in response to ECM forces, and they provide migrating cells the ability to probe the mechanics of the microenvironment[1,40,74]. However, we did not observe major induction of filopodia upon Fascin1 overexpression, Arp2/3 inhibition or CAPZ inactivation, at least under our imaging

**Fig. 4 Fascin1 is overexpressed in intrahepatic cholangiocarcinomas and required for disease progression. a** Representative Fascin1 immunohistochemistry in intrahepatic cholangiocarcinomas (iCCA) formed in FVB/N mice transduced by hydrodynamic tail vein (HTV) injection with transposon plasmids encoding for Notch Intracellular Domain (NICD), or myristoylated-AKT together with NICD (AKT/NICD), with N-Ras V12D (AKT/N-Ras), and with YAP S127A (AKT/YAP). Fascin1 immunoreactivity was limited to liver sinusoids, stromal, and endothelial cells in the normal tissue and most iCCA models, including the hepatocellular carcinomas developed in the AKT/N-Ras mice (inset). In contrast, cholangiocellular lesions developing in AKT/NICD mice exhibited intense cytoplasmic staining for Fascin1 in tumor cells. $n = 5$ mice for each model were consistent. Original magnifications: 100x; scale bar $= 100$ μm. **b** Representative IF of liver sections from mice transduced by HTV injection with HA-AKT/NICD and control short-hairpin RNA (shCo.) or with a short-hairpin RNA targeting Fascin1 (shFSCN1). Liver sections were stained for HA (see Supplementary Fig. 4B), endogenous Fascin1 and DAPI as nuclear counterstain. Livers were analyzed 3 weeks after HTV. Scale bar $= 60$ μm. **c** Quantifications of the number and size of liver lesions shown in **c**. On top: mean and single points (mice, $n = 3$) of the total number of HA-positive lesions in 5 non-consecutive sections from each liver (series), $p = 0.0008$ by unpaired Welch's t-test. Below: mean and single points of HA-positive lesions' size from $n = 3$ mice, $p < 0.0001$ by Mann–Whitney test. **d** Real-time growth curve of four different human intrahepatic cholangiocarcinoma (iCCA) cell lines treated with increasing doses of G2 Fascin inhibitor, as measured by the xCELLigence Real Time Cell Analysis system and expressed as Normalized Cell Index. **e** EdU incorporation assay of four different human iCCA cell lines transfected with control (siCo.) or validated YAP/TAZ siRNA (siY/T)[86,87], 48 h after transfection. $n = 2$ (>200 cells per condition in total). **f** Representative immunofluorescence for YAP/TAZ in KKU-156 iCCA cells transfected with control (siCo.) or Fascin1 siRNAs (siFSCN1 A and B). On the right: quantification of YAP/TAZ nuclear localization in KKU-156 and RBE cells. $n = 2$ (>50 cells per condition in total). **g** Representative Fascin1 immunohistochemistry on human iCCA. In the normal liver, Fascin1 immunoreactivity is limited to liver sinusoids and endothelial cells. iCCA case 1 shows intense cytoplasmic staining for Fascin1 in tumor cells, and this is representative of 35/62 (56%) cases. iCCA case 2 displays immunolabeling for Fascin1 in vascular structures but not in tumor cells. CK19 immunoreactivity was used to demonstrate the cholangiocellular differentiation of the iCCA lesions. Original magnification: 200x; scale bar $= 100$ μm. **h** Left panel: low magnification of a human liver section stained for Fascin1, containing a malignant preinvasive condition (black inset, intraductal papillary neoplasm of the bile duct, IPNB) and an invasive iCCA lesion (red inset) adjacent to each other. Black panel: higher magnification of the IPNB, showing immunoreactivity for Fascin1 limited to the invasion front (black arrow), but otherwise negative (white arrow). 4/13 cases were positive. Red panel: higher magnification of the iCCA, exhibiting strong and homogeneous and robust staining for Fascin1 in the cytoplasm of tumor cells. 9/13 cases were positive. Original magnification: 200x; scale bar $= 100$ μm. **i** Kaplan–Meier survival analysis of human iCCA patients based on Fascin1 expression levels. Tumors positive for Fascin1 staining have poorer outcomes (median survival 15 weeks) than negative ones (median survival 60 weeks). $n = 62$. $p = 0.0001$ by log rank Mantel–Cox test.

conditions. Moreover, Fascin1 staining in cholangiocarcinomas appears cytoplasmic diffuse and it is not limited to the cell periphery/protrusions. Thus, these players might thus regulate other cytoplasmic bundled actin structures relevant for mechanotransduction, as suggested by decreased radial bundles upon loss of Fascin1.

Importantly, this new activity of Fascin1 is relevant in the liver, a model system to study YAP and mechanotransduction. We found that Fascin1 activation is sufficient to induce mouse hepatocyte dedifferentiation into atypical ductal cells (ADC, also called oval cells or bipotent liver progenitor cells) in a YAP/TAZ-dependent manner, and that Fascin1 is required for ADC formation downstream of CAPZ inactivation, where enhanced contractility drives YAP activation[25]. These findings led us to explore the role of Fascin1 in liver cancer. We found that human intrahepatic cholangiocarcinomas (iCCA) exhibit high levels of Fascin1, promoting tumor cell proliferation, and whose expression correlates with more invasive iCCAs and poor prognosis. Moreover, we found a similar overexpression in cholangiocellular lesions induced by experimental activation of AKT and Notch in mice, providing candidate molecular mechanisms mediating Fascin1 overexpression. Regulation of Fascin1 expression observed in tumors appears different from the control of Fascin1 localization that we observe upon inhibition of CAPZ and Arp2/3 in vitro. Still, increased expression is sufficient to activate YAP and the corresponding biological responses, as we found that increased Fascin1 is required for the development of lesions induced by AKT and Notch, and it cooperates with AKT to induce cholangioma formation.

We thus propose that Fascin1 represents the first player of the YAP mechanotransduction machinery whose expression can be selected in a cell-autonomous manner by oncogenes to overcome a mechanical tumor-suppressive environment. This mechanism might be relevant in the pancreas, where stromal stiffening is a relevant driver of tumor development[75,76], and where Fascin1 is also genetically required[77]. Fascin inhibitors have been developed for use in vitro, and more recently also in vivo[47,78–80]. Since inhibition of Fascin1 is generally compatible with adult tissue homeostasis[77,81,82], Fascin inhibitors might represent an innovative approach against iCCA malignancy.

## Methods

**Cell lines, transfections, and microfabrications.** Human mammary epithelial MCF10A cells (ATCC, STR profiled, checked for endogenous mutations against other derivatives) were cultured in DMEM/F12 supplemented with 5% Horse Serum, 2 mM Glutamine, insulin (Sigma), cholera toxin (Sigma), hEGF (Peprotech) and hydrocortisone (Sigma) as in Debnath et al., 2003. Human breast cancer MDA-MB-231 cells (ATCC, STR profiled) in DMEM/F12 with 10% FBS and 2 mM Glutamine. Human intrahepatic cholangiocarcinoma cell lines KKU-156, KKU-213 (STR profiled) in RPMI1640 with 10% FBS, 10 mM HEPES, 2 mM Glutamine, and 1 mM Sodium Pyruvate, RBE and HUCCT1 (STR profiled) in DMEM high glucose with 10% FBS, 10 mM HEPES, 2 mM Glutamine, and 1 mM Sodium Pyruvate. Murine breast cancer E0771 cells were cultured in RPMI medium supplemented with 10% FBS, 2 mM Glutamine, and 1% HEPES. All cell lines were routinely tested with universal mycoplasma ATCC detection kit 30-1012 K and were negative. siRNA transfections were done with Lipofectamine RNAi MAX (Invitrogen) and plasmid DNA transfections were done with Transit-LT1 (MirusBio) according to the manufacturer's instructions. Where indicated, transfections were carried out on plastic vessels and cells were subsequently replated on hydrogels. Fibronectin-coated polyacrylamide hydrogels ($E = 0.5$ kPa) were assembled in-house as in ref.[25].

**Reagents.** The small-molecule inhibitors were SMIFH2 (Sigma s4826, 60 microM), CK-869 (Sigma C9124, 80 microM) LatrunculinA (Sigma L5163, 0.5 microM), G2 (Xcessbio M60269 for cell treatments, Valuetech custom synthesis for mouse injections). All plasmids were sequence-verified before use and transfected as endotoxin-free maxi preps. siRNAs were selected among FlexiTube GeneSolution 4 siRNA sets (Qiagen) and reordered after validation as dTdT-overhanging 19 nt RNA duplexes (Thermo). shRNAs were selected among pre-validated Mission pLKO1-shRNA (Sigma) and the corresponding U6-shRNA-cPPT cassettes were subcloned into PB-empty vector[25]. Sequences of siRNAs are provided in Table 1.

**Human tissue samples.** Sixty-two intrahepatic cholangiocarcinomas (iCCA) and fifty hepatocellular carcinomas (HCC) as well as the corresponding surrounding non-tumorous liver tissues were used for the study. Patients' clinicopathological features are summarized in Supplementary Tables 1 and 2. Tumors were divided in iCCA/HCC with shorter/poorer (iCCA, $n = 35$; HCC, $n = 28$) and longer/better (iCCA, $n = 27$; HCC, $n = 28$) survival, characterized by <3 and ≥3 years' survival following partial liver resection, respectively. Liver tissues were collected at the Universities of Greifswald (Greifswald, Germany) and Regensburg (Regensburg, Germany). Institutional Review Board approval was obtained at the local Ethical Committees of the Medical Universities of Greifswald (approval code: BB 67/10) and Regensburg (17-1015-101), in compliance with the Helsinki Declaration. Written informed consent was obtained from all individuals.

**Mice and treatments.** Mice were C57BL/6 N as in ref.[25]. Sex allocation was random. For G2 treatment, mice were administered 100 mg/kg of G2 by daily i.p.

**Table 1 siRNA and shRNA sequences.**

| Target gene | siRNA or shRNA Sequence |
|---|---|
| hFSCN1 A | UGGCAAGUUUGUGACCUCC |
| hFSCN1 B | CAGCGUCACCCGUAAGCGC |
| hCAPZB A | GAAGUACGCUGAACGAGAUck |
| hCAPZB B | GGAGUGAUCCUCAUAAAGA |
| AllStars negative control (Qiagen) | Not available—proprietary information |
| Scramble control shRNA | UUCUCCGAACGUGUCACGU |
| mFSCN1 A | CCGGGCATCCGCTAGTAGCTTGAAACTCGAGTTTCAAGC TACTAGCGGATGCTTTTTG |
| mFSCN1 B | CCGGCCTCCTGTTATCCTTACTCATCTCGAGATGAGTAAG GATAACAGGAGGTTTTTG |

**Table 2 Antibodies.**

| Epitope | Brand and catalog | Validation |
|---|---|---|
| CK19 | DSHB TROMAIII | Positive IF staining on bile ducts. NOTE: the antibody requires inclusion of fresh liver tissues in OCT without prior fixation with PFA. |
| A6 | DSHB A6 BCM | Positive IF staining on bile ducts. |
| YAP/TAZ | SC-101199 used in vitro | Detects a prominent band in WB of the expected size, which fades in human cells treated with YAP siRNA. Detects nuclear/cytoplasmic shuttling of YAP in multiple conditions. |
| YAP | PROTEINTECH 13584-1-AP used in vivo | Detects a prominent band in WB of the expected size, which fades in human cells treated with YAP siRNA. Detects nuclear/cytoplasmic shuttling of YAP in multiple conditions in vitro. IF signal is lost in YAP-null liver tissue. |
| FSCN1 | SC-21743 | IF staining shows a specific localization in filopodia and in cytoplasm colocalized with phalloidin. The staining fades in human cells and mouse tumors expressing FSCN1 siRNAs or shRNAs. |
| FSCN1 | ab126772 | Staining is consistent with the other antibody used in the study |
| HA | SC-7392 | Positive WB or IF staining only in cells transduced with plasmids encoding for HA-tagged proteins |
| β-gal | ab9361 | Positive IF staining only in tissues or cells expressing a lacZ transgene. |

injection, 5 days per week, for 2 weeks, starting together with tamoxifen injection (to induce CAPZB recombination). Animal experiments were performed according to our institutional guidelines as approved by the University Animal Welfare Commission (OPBA) and authorized by the Ministry of Health (945/2015-PR and 54/2015-PR). Reporting was according to the ARRIVE guidelines.

Hydrodynamic tail-vein injection was used to transduce hepatocytes of 4/6-week-old mice with exogenous DNA. 50 μg of total PiggyBac (PB) and/or Sleeping Beauty (SB)-transposon plasmid DNA together with 10 μg of hyperactive PB Transposase plasmid DNA (hyPBase) or hyperactive Sleeping Beauty transposase (pCMVT7-SB100, Addgene 34879) were diluted in sterile Ringer's solution in a volume corresponding to 10% of the bodyweight (on average, 18–22 g), and injected via the tail vein in a maximum time of 8–10 s. PB-transposon plasmids were obtained by subcloning the cDNAs of interest (GFP, Fascin1 S39A-IRES-GFP, FLAG-mTAZ 4SA) in a PB-empty plasmid.

Mice were euthanized and abdominal contents exposed. Trans-cardiac perfusion (29-gauge needle) with cold 1XPBS (10–20 ml) was performed to reduce blood contaminants. The liver was removed and placed in a clean petri dish with 1XPBS on ice. The liver was immediately divided into parts and snap-frozen in liquid nitrogen for extraction of mRNA/proteins or embedded in OCT and stored at −80° for subsequent analyses.

**Immunofluorescence.** For immunofluorescence, we followed refs. [83–85] with minor modifications. Cells were fixed in 4% PFA for 10 min. After washes, cells were permeabilized with PBS-Triton 0.5% for 20 min following by blocking buffer (PBS- Triton 0.1%, 2% goat serum) incubation for 1 h. The primary antibody (see Table 2 below) was incubated overnight at +4°. The day after, cells were incubated with AlexaFluor-conjugated secondary antibodies (Invitrogen) for 1 h at room temperature. Dishes were mounted with ProLong™ Gold Antifade Mountant with DAPI (P36935 Thermofisher). Images were acquired with a Leica SP5 or with a ZEISS LSM700 confocal microscope equipped with a CCD camera, using Leica LAS AF or ZEN 2 software, or with a standard Leica DM5000B microscope. Raw images were analyzed with ImageJ (Fiji).

For immunofluorescence on liver sections, OCT-embedded tissue was cut into 5–8 μm thick sections with a Leica CM1950 cryostat. Sections were dried at RT for 30 min on a glass coverslip (VWR), and either stored dried at −80 °C or directly processed by rehydration in 1XPBS followed by fixation in 4% PFA for 15 min. Permeabilization was performed in 1XPBS-Triton 1% for 20 min. Blocking was 10% goat serum in 1XPBS-Triton 0.5% for 1 h at RT.

For EdU labeling, mice were injected with 12.5 mg/kg of EdU in sterile 1XPBS (A10044 Molecular Probes) 15 h before tissue sampling. Cells were incubated for 1 h with EdU in sterile 1XPBS. Liver slice or cells were fixed in PFA 4% and block/permeabilize for 30 min in 1xPBS 3% BSA + 0.2% Triton (1% Triton for liver slices). EdU reaction mix (100 mM Tris pH 8.5, 4 mM CuSO4, 625 nM Alexa Azide, 100 mM Ascorbic acid) was incubated for 30 min.

**Histology and immunohistochemistry.** Human and mouse liver specimens were fixed overnight in 4% paraformaldehyde and embedded in paraffin. Tumors arising in mice from different oncogene combinations and used for Fascin1 IHC were induced in FVB/N females (see references in the pertaining section of text). Sections were done at 5 μm in thickness. Liver lesions were evaluated and classified by two board-certified pathologists and liver experts (S.R. and M.E.). For immunohistochemistry, slides were deparaffinized in xylene, rehydrated through a graded alcohol series, and rinsed in PBS. Antigen unmasking was achieved by boiling in 10 mM sodium citrate buffer (pH 6.0) for 10 min, followed by a 20-min cool down at room temperature. After a blocking step with the 5% goat serum and Avidin-Biotin blocking kit (Vector Laboratories, Burlingame, CA), human tissue slides were incubated with primary antibody overnight at 4 °C (see Table 2). Slides were subjected to 3% hydrogen peroxide for 10 min to quench endogenous peroxidase activity and, subsequently, the biotin conjugated secondary antibody was applied at a 1:500 dilution for 30 min at room temperature. Immunoreactivity was visualized with the Vectastain Elite ABC kit (Vector Laboratories, Burlingame, CA), using Vector NovaRed (Vector Laboratories) as the chromogen. Slides were counterstained with hematoxylin.

**Luciferase assays.** Cells were plated in 24-well plates and transfected with YAP/TAZ luciferase reporter 8XGTIIC-lux plasmid (50 ng/cm2) (Addgene 34615) together with CMV-lacZ (75 ng/cm²) to normalize for transfection efficiency based on CPRG (Merck) colorimetric assay. Transfected DNA content was kept equal using pKS Bluescript. Cells were harvested in luc lysis buffer (25 mM Tris pH 7.8, 2.5 mM EDTA, 10% glycerol, 1% NP-40). Luciferase activity was determined in a Tecan plate luminometer with freshly reconstituted assay reagent (0.5 mM D-Luciferin, 20 mM tricine, 1 mM (MgCO3)4Mg(OH)2, 2.7 mM MgSO4, 0.1 mM EDTA, 33 mM DTT, 0.27 mM CoA, 0.53 mM ATP). Each sample was transfected in two biological duplicates; each experiment was repeated independently with consistent results.

**RNA extraction and gene expression studies.** Total RNA was isolated using commercial kits with DNAse treatment (Norgen). cDNA synthesis was carried out

**Table 3 Primer sequences.**

| qPCR Primer | For | Rev |
|---|---|---|
| hGAPDH | CTCCTGCACCACCAACTGCT | GGGCCATCCACAGTCTTCTG |
| hFSCN1 | CAAGAAGAATGGGCAGCTGG | CTTTGATGTTGTAGGCGCCA |
| hFSCN2_1 | ACGAGACCTTCCTGATGCAA | CCAGCTGCCCATTCTTCTTC |
| hFSCN2_2 | GAAGAAGAATGGGCAGCTGG | CAGGTGGAAGACGTCGTAGA |
| hFSCN3_1 | TCCAGGCCCAAATGAGGAAT | CTGTGCCTGGAAGTGGTAGA |
| hFSCN3_2 | GCGCTTAAACCGAATGTCCT | ATGATCCTGCCACAGTTCCA |
| mGAPDH | ATCCTGCACCACCAACTGCT | GGGCCATCCACAGTCTTCTG |
| mANKRD1 | CTGTGAGGCTGAACCGCTAT | TCTCCTTGAGGCTGTCGAAT |
| mCTGF | CTGCCTACCGACTGGAAGAC | CATTGGTAACTCGGGTGGAG |
| mFSCN1 | CAAGTTTGTGACCGCCAAGA | GTAGGCGCCGTCATTGAATT |

with M-MLV Reverse Transcriptase (Thermo) and oligo-dT primers. qPCR reactions were assembled with FastStart SYBR Green Master Mix (Roche) and run on a QuantStudio6 thermal cycler (Thermo). Gene expression levels for each biological sample were quantified as the mean between three technical replicates; *GAPDH* expression levels were used to normalize gene expression between samples. See Table 3 for primer sequences.

**Statistics and reproducibility**. Sample size was determined based on previous experience. All data consist of independent experiments with independent biological replicates. All *n* values are pooled between independent experiments. Samples were not blinded for analyses. Data analyses were performed using GraphPad Prism software. Graphs indicate mean values and single points of all biological replicates (or mice) unless otherwise indicated. Data for each mouse is the average of multiple ($n \geq 6$) randomly selected non-adjacent tissue sections. Significance was calculated by applying unpaired Welch's *t*-tests between the indicated samples unless indicated otherwise in the figure legends.

**Reporting summary**. Further information on research design is available in the Nature Research Reporting Summary linked to this article.

## Data availability

Source data for the figures are available in the Supplementary Data 1 file and any remaining information can be obtained upon reasonable request to the corresponding author.

## Code availability

No custom generated codes were used.

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

## Acknowledgements

We are indebted to Giorgio Scita for pointing us towards the CAPZ vs. Ena/VASP antagonism, and for sharing the A4P-GFP-mito and F4P-GFP-mito plasmids; Nils Gauthier for the mCherry-Vinculin plasmid; Graziano Martello for the hyPBase plasmid; Duoja Pan for the Yap1fl/fl mice; Stefano Piccolo for the Wwtr1(TAZ)fl/fl mice; Antonio Rosato for help establishing the HTV technique; Anna Mondino for generous help during the revision. This work was supported by an AIRC Foundation Investigator Grant 21392, a CARIPARO Foundation Eccellenza 2017 Grant and University of Padova BIRD Grant to SD. Work in the SD lab is also supported by a Worldwide Cancer Research grant 21-0156. A.P. was the recipient of an IFOM/AIRC Post-Doctoral fellowship. P.R. is the recipient of a Veronesi Foundation Post-Doctoral fellowship. Work in the XC lab was supported by NIH grants R01CA190606 to XC, and P30DK026743 for UCSF Liver Center.

## Author contributions

A.P. performed in vitro and in vivo experiments, analyzed data, assembled figures, and wrote the first draft. G.S. and P.R. helped to produce data during the revision. C.A. acquired and analyzed clinical data. S.R., X.C., and M.E. provided clinical material, technical (pathologist), and related funding support. D.F.C. was involved in clinical study design and study supervision. S.D. coordinated and supervised the study, provided funding, and wrote the manuscript.

## Competing interests

The authors declare no competing interests.
