## [Peer Review File · Communications Biology]

Reviewers' comments:

Reviewer #2 (Remarks to the Author):

In this study by Pocaterra et al., the authors performed a series of experiments to identify molecules that are important for YAP-mediated mechanotransduction by taking advantage of the previously published work that established loss of actin capping proteins as an inducer for mechanotransduction-dependent activation of YAP/TAZ. They discovered the actin bundling protein Fascin1 as an important factor in YAP/TAZ activation both in vitro and in vivo. They showed that knockdown or inhibition of Fascin1 causes YAP cytoplasmic localization and lower YAP/TAZ activity in cells plated on a stiff substrate while ectopic Fascin overexpression induces YAP nuclear localization and activity in cells plated on soft substrate. They also showed that inhibition of Arp2/3 pathway also induces YAP nuclear localization and this process also requires Fascin1. Moreover, they found that Fascin1 regulates YAP/TAZ-dependent hepatocyte dedifferentiation into atypical ductal cells. Finally, they provided evidence on the importance of Fascin1 in intrahepatic cholangiocarcinoma development in mice and cholangiocarcinoma disease progression in humans.

This manuscript is interesting, given that it provides a biologically relevant example to mechanotransduction-mediated YAP activation. It is very well-written and most of the results are elegant and clear. However, the following points should be addressed in the manuscript for it to be published in Communications Biology.

Major points:

1. The authors suggest that the actin bundling activity of Fascin is responsible for its effect on YAP/TAZ without providing any evidence supporting this view. Does overexpression of Fascin1 induce bundled F-actin? The authors should do an experiment to test whether F-actin is required for Fascin1 to activate YAP, for example in cultured cells by adding Latrunculin. Also, what is the effect of Fascin1 kd in the cells tested?
2. Extrapolating from the previous point, there is no experimental evidence suggesting that Fascin1-dependent actin bundling is specifically important but not any form of actin bundling. What makes Fascin1-dependent actin bundling special for YAP activation in the context of mechanotransduction? This should be mentioned in the discussion.
3. The authors generally used the S39A mutant Fascin1 in their overexpression assays. They should mention the effect of this mutation. Does it affect functions of Fascin1 other than actin bundling? Also, does overexpression of the wild-type protein have similar effects?
4. In Fig2C, it is not clear or convincing enough from the pictures to suggest Fascin1 is enhanced upon CPAZB knockdown. Is it possible to show this quantitatively?
5. Fig2F, These pictures are difficult to interpret. The nuclei are not easily delineated from the composite pictures. Maybe it would be better to also show the nuclei only in a separate panel and maybe add a staining to show cell outlines. From the pictures shown it seems that the control has more nuclear YAP than the Fascin1 overexpression.
6. The effect in Fig3A is weak, which is acknowledged in the manuscript. Is it possible to see the image of a larger piece of the section?
7. Fig3B is problematic. First, DAPI channel should be added (and shown in a separate panel as dark blue is barely visible in a composite image). Second, GFP staining looks very different in the upper panel than that of lower panel. Third, Edu as a readout here is not convincing at all. They see low levels in normal livers but barely any change in the mutant condition. Forth, what is the meaning of EdU positive cell percentage here? Is it the percentage of EdU positive cells in GFP positive cells or is it the EdU positive cells in the entire liver section?
8. Fig3B, what is the percentage of transfected cells that induce A6 expression?

9. Fig3, most importantly, they should look at YAP localization.
10. Fig3C, this requires quantification and also inclusion of the wt control!
11. Fig3D and E, requires pictures of the livers and H&E sections (large view and close up). Also, what about differences in tumor size?
12. Fig4, Fascin1 knockdown cells do not established tumors in NICD-Akt model, are those cells still cancer cells? Are they positive for CK19 or A6?
13. What is the phenotype of loss of Fascin1 in vitro and in vivo? How does that effect a potential therapy based on Fascin1 inhibition?

Minor points:

- In Figure 1F, please indicate the different doses of G2 on the graph itself to differentiate which one is which.
- In Fig2F, Fascin staining should be added. Showing only GFP creates a confusion that GFP tagged protein was expressed but overexpressed protein is not tagged (It is GFP-IRES-Fascin).
- Graph representation in Sup Fig4 needs improvement. It is impossible to read the labels on both axes. Also, are these cell lines YAP-dependent? It might be good to see a control related to YAP here.

Reviewer #3 (Remarks to the Author):

The manuscript by Pocaterra et al., describes a novel function for Fascin1 in YAP mechano transduction. The authors start by characterizing the role of proteins such as the formins, Ena/VASP and Fascin in regulation of YAP. Significantly, the authors show that inhibition of Fascin by knockdown or small molecule leads to accumulation of YAP/TAZ in the cytoplasm and reduced transcriptional activity. In addition, they demonstrate that inhibition of Arp2/3 reduces YAP/TAZ nuclear localization, which phenocopies CAPZ depletion. Moreover, this appears to be mediated through Fascin1. The authors were able to corroborate some of their findings in vivo, by overexpressing Fascin1 in the liver and demonstrating this effect requires YAP/TAZ. In support of the function of Fascin1 as an activator of YAP/TAZ, the expression of Fascin1 along with activated AKT in the mouse liver leads to development of CCA. Using different genetic combinations to induce CCA, the authors observe Fascin1 upregulation in the myrAKT/NICD model. This is of interest since this combination relies on YAP for induction of CCA. Indeed, inhibition of Fascin1 in this model inhibited CCA development. Finally, the authors interrogated the expression of Fascin1 in liver tumors from patients and find increased levels in CCA and to a lesser extent in HCC.

This is an interesting report that is appropriately controlled. It provides interesting observations regarding the mechanisms by which mechanotransduction regulates signaling through YAP and provides a relevant in vivo context. My suggestions are minor and listed below:

1. The effects of G2 treatment in vivo, as shown in Figure 3A, are subtle. One contributing factor could be the IP method of delivery. Since the authors have experience with HDTV injection, it might be worthwhile trying this as a method of delivery to enhance the impact of G2 treatment.
2. While this might be beyond the scope of the manuscript, it would be interesting to assess the function of Fascin1 in the same cell type grown on substrates that have a stiffness gradient.

Point-by-point response to Reviewers' comments:

Reviewer #2 (Remarks to the Author):

In this study by Pocaterra et al., the authors performed a series of experiments to identify molecules that are important for YAP-mediated mechanotransduction by taking advantage of the previously published work that established loss of actin capping proteins as an inducer for mechanotransduction-dependent activation of YAP/TAZ. They discovered the actin bundling protein Fascin1 as an important factor in YAP/TAZ activation both in vitro and in vivo. They showed that knockdown or inhibition of Fascin1 causes YAP cytoplasmic localization and lower YAP/TAZ activity in cells plated on a stiff substrate while ectopic Fascin overexpression induces YAP nuclear localization and activity in cells plated on soft substrate. They also showed that inhibition of Arp2/3 pathway also induces YAP nuclear localization and this process also requires Fascin1. Moreover, they found that Fascin1 regulates YAP/TAZ-dependent hepatocyte dedifferentiation into atypical ductal cells. Finally, they provided evidence on the importance of Fascin1 in intrahepatic cholangiocarcinoma development in mice and cholangiocarcinoma disease progression in humans.

This manuscript is interesting, given that it provides a biologically relevant example to mechanotransduction-mediated YAP activation. It is very well-written and most of the results are elegant and clear.

We thank the Reviewer for the positive and constructive comments. As detailed below, we worked along the lines indicated by this Reviewer. We now provide more controls, better pictures and quantifications. Moreover, we also provide new experiments and analyses that we believe reinforce the manuscript. We hope the Reviewer will now find the manuscript suitable for publication, also considering how important it is in these tough times to comply with the needs of granting agencies and to provide young scientists with opportunities for their future (of which publications are at the very core).

However, the following points should be addressed in the manuscript for it to be published in Communications Biology.

Major points:

1. The authors suggest that the actin bundling activity of Fascin is responsible for its effect on YAP/TAZ without providing any evidence supporting this view. Does overexpression of Fascin1 induce bundled F-actin? The authors should do an experiment to test whether F-actin is required for Fascin1 to activate YAP, for example in cultured cells by adding Latrunculin. Also, what is the effect of Fascin1 kd in the cells tested?

Thanks for the suggestion. We performed the requested experiment, and now show that overexpression of Fascin1 is not able to favor YAP/TAZ nuclear translocation in cells treated with latrunculinA (new panels in Supplementary Fig. 2F). This supports the view that Fascin1 regulates YAP/TAZ by regulating F-actin.

In response to the specific questions on F-actin, we did not observe formation of macroscopic bundle structures in cells overexpressing Fascin1. This might also be due to the limited imaging tools we have at our disposal. This is however a very general question left unaddressed in the field, namely what F-actin structures are formed in response to differential ECM stiffness - we know much about plastics, but very little about more physiological stiffnesses. As for Fascin inhibition or kd, we now provide more pictures (Supplementary Fig. 1C) and comment on how this

mainly affects formation of radial bundles (line 126), while circumferential ones persist to some extent. However, it must be considered that Fascin1 contributes to multiple F-actin structures in cells (see References in the text and discussion), and it is currently not possible to alter these different structures selectively to know which is more relevant. We included a discussion on filopodia to underline this point (lines 293-303), which now ends with:

“Thus, these players might regulate other cytoplasmic bundled actin structures relevant for mechanotransduction, as suggested by decreased radial bundles upon loss of Fascin1.”

Legend: Representative immunofluorescence images of MCF10A cells transfected with GFP or with GFP+FSCN1 plasmids and treated with latrunculinA for 6 hours. Scale bar= 4µm

Legend: Representative immunofluorescence images of MCF10A cells transfected with FSCN1 siRNA (siFSCN1) or control siRNA (siCo.) and stained for Fascin1, F-actin (phalloidin) and DAPI. Asterisks indicate radial Fascin1-positive F-actin bundles. Scale bar= 10µm.

2. Extrapolating from the previous point, there is no experimental evidence suggesting that Fascin1-dependent actin bundling is specifically important but not any form of actin bundling. What makes Fascin1-dependent actin bundling special for YAP activation in the context of mechanotransduction? This should be mentioned in the discussion.

Thanks for pointing this out. We focused on Fascin1 because of previous data on CAPZ/YAP mechanotransduction and CAPZ/Ena/Fascin1 antagonism, but our data do not exclude the possibility that multiple actin-bundling proteins could equally empower YAP mechanotransduction. We now explicitly mention this additional possibility in the discussion (lines 289-292):

“...in these conditions, not only RHO signalling is decreased, leading to decreased activity of ROCK/MLCK and of Formins (1), but at the same time CAPZ-dependent and Arp2/3-dependent actin outbalances actin bundles promoted by Ena/VASP, Fascin1 and potentially other F-actin bundling proteins.”

3. The authors generally used the S39A mutant Fascin1 in their overexpression assays. They should mention the effect of this mutation. Does it affect functions of Fascin1 other than actin bundling? Also, does overexpression of the wild-type protein have similar effects?

We introduced the complete description at line 164, together with the original reference on the S39A mutant. The S39A mutation is believed to abolish an inhibitory phosphorylation site. We used this from the beginning of our experiments as we wanted to use a FSCN1 isoform capable of overcoming potential cross-talks and feed-backs, which are commonly observed in many processes. We are unaware whether the mutation affects other Fascin1 functions. To our knowledge, this remains uncharacterized also in the published literature.

We performed an experiment in HEK293 cells, a system easily amenable to transient transfection and displaying low levels of endogenous YAP/TAZ activity, where both WT and the S39A mutant Fascin1 can increase the 8xGTIIC-lux YAP/TAZ reporter activity at high density (Supplementary Fig. 2E).

Legend: Luciferase assay with the 8XGTIIC-lux YAP/TAZ reporter in HEK293 cells at high density. Co. indicates cells transfected only with the reporters and filler DNA. Mean expression levels in the Co. sample were set to 1, and all other samples are expressed relative to this. n=2 experiments. Data are mean and s.d.

4. In Fig2C, it is not clear or convincing enough from the pictures to suggest Fascin1 is enhanced upon CPAZB knockdown. Is it possible to show this quantitatively?

We now provide a quantification of the stainings in Fig. 2C.

Legend: Representative immunofluorescence images of MCF10A transfected with CAPZB siRNA (siCAPZB) or control siRNA (siCo.) and replated on soft hydrogels for 48 hours, stained for Fascin1 and F-actin (phalloidin). Quantification of Fascin1 intensity on the right. n=2 independent experiments. Scale bar= 2 μ m

5. Fig2F, These pictures are difficult to interpret. The nuclei are not easily delineated from the composite pictures. Maybe it would be better to also show the nuclei only in a separate panel and maybe add a staining to show cell outlines. From the pictures shown it seems that the control has more nuclear YAP than the Fascin1 overexpression.

We are sorry for that, the dark-blue DAPI channel did not display well. We now provide the requested panels (new Fig. 2F), and added the outline of the nuclei in the YAP/TAZ channel to make clear what is the relevant part of the picture.

6. The effect in Fig3A is weak, which is acknowledged in the manuscript. Is it possible to see the image of a larger piece of the section?

As mentioned in the original Methods, these quantifications were made based on 4 randomly chosen pictures of periportal areas in each of 4 different sections of every mouse under analysis. The images were acquired at a magnification low enough to analyze a large area, but high enough to provide a good signal for quantification. We now provide in Supplementary Figure 3 additional images so that it is possible to gauge multiple sections, thus covering larger pieces. These (and others) were originally used for the quantification provided in Fig. 3A.

The weak effect could be due to the use of a compound that has not been optimized for in vivo experiments. Anyway, the effect is in line with a modulatory role of Fascin1 on YAP activity, and with the effects observed in cell culture (where Fascin1 inhibition has weaker effects than latrunculinA, for example), thus supporting our findings.

7. Fig3B is problematic. First, DAPI channel should be added (and shown in a separate panel as dark blue is barely visible in a composite image). Second, GFP staining looks very different in the upper panel than that of lower panel. Third, Edu as a readout here is not convincing at all. They see low levels in normal livers but barely any change in the mutant condition. Fourth, what is the meaning of EdU positive cell percentage here? Is it the percentage of EdU positive cells in GFP positive cells or is it the EdU positive cells in the entire liver section?

Thanks for pointing this out. The old Fig3B has now been split in the new Fig. 3B and 3C. Moreover, we now provide new pictures and added separate channels so that it is possible to more easily understand each staining. In the new Fig. 3B we also included the control with hepatocytes expressing only GFP, so that it is easier to visually follow the experiment. We hope these are now clear examples of the data, for which we had already provided the relevant quantifications in the original panels.

As for EdU, we calculated the percentage of EdU positive cells only in GFP-positive cells, because in hydrodynamic tail vein (HTV) injection experiments not all cells are targeted, and only those positive for GFP are relevant to the analysis. This is now clearly stated also in the figure legend.

The basal levels of proliferation in control GFP+ hepatocyte is similar to that observed in non-transduced hepatocytes and is in line with previous studies from our lab and from others - please note these are livers from adult mice, where proliferation is expected to be lower than in the postnatal period. Still, we observed twice as much proliferation in hepatocytes transduced with FSCN1. This is weaker compared to CAPZ or LATS1/2 inactivation in a similar HTV setting, reaching a 3- to 4-fold increased proliferation (Pocaterra JHEP 2019). However, it must be noted these stronger effects happened on a longer time-scale (one month for CAPZ and LATS1/2 vs. two weeks for FSCN1). Then, obviously, the effects of FSCN1 are also weaker compared to the original knockouts of the Hippo pathway. This is also expected, because these inactivated major regulators of YAP/TAZ (i.e. LATS1/2, MST1/2, SAV), and also because these were constitutive knockouts where recombination occurred from the early phases of foetal hepatocyte development. Thus, we believe that even showing a two-fold increase in proliferation in adult hepatocytes, coupled with other phenotypes such as dedifferentiation, indicates FSCN1 as a relevant regulator of YAP activity in vivo.

E

Legends: B. Representative immunofluorescence stainings of liver sections from C57BL/6N mice transduced by hydrodynamic tail vein (HTV) injection with transposon plasmids encoding for GFP or for GFP+Fascin1. Liver sections were stained for EdU. Analyses were carried out 15 days after HTV injection. On the right, quantification of EdU-positive GFP-positive hepatocytes. Mean and single points (mice, n=3). Scale bar = 5µm. C. Representative immunofluorescence stainings of liver sections from mice transduced by HTV with transposon plasmids encoding for GFP or GFP+Fascin1as indicated. YAP/TAZ KO was obtained by adding the CRE recombinase to the HTV mix, and by injecting *Yap1^{fl/fl}*; *Wwtr1(TAZ)^{fl/fl}*; *ROSA26-LSL-lacZ* mice. Analyses were carried out 15 days after HTV injection and induction of CRE activity by tamoxifen. Recombination was controlled by β-galactosidase staining. Mean and single points (mice, n=3). Scale bar= 10µm.

8. Fig3B, what is the percentage of transfected cells that induce A6 expression?

We now provide this data in Fig 3C (see above). In this panel, we also included all the relevant conditions (GFP, GFP+FSCN1 HTV in wt mice, and GFP+FSCN1 HTV in YAP/TAZ KO mice), so that it is easier to visually follow the experiments. In this case, this is an all-to-nothing effect, because expression of GFP by HTV in hepatocytes never induced A6 positivity in our hands.

9. Fig3, most importantly, they should look at YAP localization.

We tried this experiment, but unfortunately did not obtain stainings of sufficient quality. The livers were originally included in OCT without prior fixation because some key stainings (e.g. CK19) are not compatible with PFA. However, this apparently hampered the quality of the YAP staining. We could not repeat the in vivo experiments due to current limitations in the use of the mouse facility related to the pandemics. We apologize for this. However, please note that we had performed experiments on YAP/TAZ knockout mice specifically to address the issue of whether the phenotypes were due to activation of YAP/TAZ, as opposed to other possible mechanisms.

As additional data, we performed new in vitro experiments on cell monolayers, which approximate to some extent the situation in a real epithelium. The formation of confluent monolayers entails the reduction of cell-ECM adhesion area and the activation of what we called a “mechanical checkpoint” similar to that occurring on a soft ECM, such that inactivation of CAPZ can rescues YAP activity in both conditions (Aragona Cell 2013; Benham-Pyle Science 2015). Nicely, we now show the same can happen by expression of FSCN1 in small cell “clones” surrounded by wild-type cells, which is

sufficient to increase YAP nuclear localization (Supplementary Fig. 2G, described at lines 166-174). Moreover, in the same experiment FSCN1 can also lead to increased proliferation in combination with oncogenic activation (Supplementary Fig. 2G). This is slightly different from the liver, where FSCN1 is sufficient to induce proliferation, which might be related to cell-type differences or to different time-frames.

Legend: Representative immunofluorescence images of MCF10A cells transiently transfected with the indicated combinations of plasmids encoding for GFP, FSCN1, H-RAS G12V, and myristoylated AKT were seeded together with non-transfected cells at high density. After 48 hours, cells were harvested for analysis of YAP/TAZ localization and EdU incorporation (quantifications on the right). Dotted white lines indicate transfected cells. YAP/TAZ and EdU levels were similar in GFP and in surrounding non-transfected cells. Data are mean and single points. n=2 experiments.

10. Fig3C, this requires quantification and also inclusion of the wt control!

We included the requested data in the new Fig. 3C. See comments above to point 7.

11. Fig3D and E, requires pictures of the livers and H&E sections (large view and close up). Also, what about differences in tumor size?

We performed in full the requested experiments (described at lines 203-215). We now provide:

- a) pictures of whole livers with HTV of AKT, AKT+FSCN1 and with HTV of AKT+TAZ, now in Fig. 3D. AKT alone had no effects in these mice (as also shown in the original manuscript by absence of anti-HA-AKT positive cells/lesions - see Fig. 3F), likely because of the B6 strain genetic background. Indeed previous data on AKT ability to induce liver tumors in mice were obtained by using the sensitized FVB strain (see References in the text from our collaborators, Dr. Calvisi and Chen). Lesions formed by AKT+FSCN1 were clearly visible on the surface of the organ. Lesions formed by AKT+FSCN1 are smaller than those formed by AKT+TAZ, but this is expected because we used the TAZ 4SA constitutively active mutant which has very powerful activity.
- b) H&E stainings of liver sections, now in Fig. 3E. AKT does not induce any type of detectable lesion on its own. AKT+FSCN1 induces smaller lesions than AKT+TAZ, in keeping with the whole liver results.

To our surprise, pathological assessment of these sections by our collaborator Dr. Calvisi indicated the formation of mixed lesions containing hepatocellular adenomas or hepatocellular foci (CK19 negative) associated with cholangiomas or neoductular proliferation (CK19 positive) - now also presented in Fig. 3E. Although TAZ 4SA induced larger lesions than FSCN1, it induced the same type of mixed lesions, and we did not notice signs of malignant progression either. We concluded that, from the point of view of the genotype/phenotype correlation, AKT is insufficient to cause a lesion, expression of FSCN1 causes the appearance of precancerous lesions, and these closely recapitulate the effects of expressing TAZ, albeit with reduced strength. This supports our original data and claims.

We note here how expression of AKT and AKT+YAP in FVB mice provides results that are slightly different: AKT induces mixed lesions on its own, while AKT+YAP only CK19-positive lesions, and with a much more accelerated progression towards malignancy. We speculate these differences are compatible with a global shift in the potency of the effects in FVB mice compared to B6 mice, which affects both the threshold to form a lesion and the propensity towards cholangiocyte dedifferentiation.

Legends: D. Representative pictures and quantification of the macroscopic lesions detected at the liver surface in C57BL/6N mice transduced by HTV injection with transposon plasmids expressing myristoylated-HA-AKT alone, together with Fascin1 (AKT/FSCN1), or together with activated TAZ 4SA (AKT/TAZ). Livers were analyzed 7 months after HTV. Mean and single points (mice, n=3). E. Representative Hematoxylin and Eosin (H&E) and immunohistochemistry for the cholangiocellular marker CK19 on liver sections from C57BL/6N mice transduced by HTV injection as in D. The black arrow indicates a hepatocellular adenoma. Squares and insets: higher magnification on CK19-positive areas, including cholangiomas and ductular reactions. Livers were analyzed 7 months after HTV. F. Representative immunofluorescence images of liver sections from C57BL/6N mice transduced by HTV injection as in D. Liver sections were

stained for HA to localize AKT-expressing cells, the cholangiocellular marker A6, and DAPI. The white arrow indicates a normal A6-positive bile duct. Above the pictures: quantification of EdU stainings in HA-positive cells. Cells expressing only AKT were undetectable, likely due to the high resistance to oncogenic transformation of the C57BL/6N strain. EdU incorporation in this condition refers to normal hepatocytes. Livers were analyzed 7 months after HTV. Average and single points (mice, n=3). Scale bar= 80µm.

12. Fig4, Fascin1 knockdown cells do not established tumors in NICD-Akt model, are those cells still cancer cells? Are they positive for CK19 or A6?

As indicated in the text, we do not think the experiments performed in Fig. 4 show real tumors but rather early neoplastic lesions. Please consider these mice are of the B6 strain, and that we analyzed the mice two weeks after HTV, thus recapitulating the early onset of the tumorigenic process. We performed the requested stainings but we did not see overt positivity for CK19 at these early time-points. However, we note how the kinetics of CK19-positivity in these lesions is unknown, and that this system is the most widely accepted model for cholangiocarcinoma formation in mice, including the B6 strain where tumors with the appropriate morphology are seen at 7 weeks after HTV (Moya Science 2019). We could not perform these control experiments at later time-points due to animal housing restrictions under the pandemics, but we used exactly the same plasmid reagents used in Moya Science 2019 and in several other publications, providing the expected results. Moreover, the increased Fascin1 expression observed upon AKT+NICD expression both in the FVB strain (Fig. 4A) and B6 strain (Fig. 4B) suggests a coherent and conserved mechanism.

13. What is the phenotype of loss of Fascin1 in vitro and in vivo? How does that effect a potential therapy based on Fascin1 inhibition?

Complete inactivation of Fascin1 is compatible with embryonic development and adult life (Yamakita 2009; Ma 2013; Li 2014) such that it was possible to perform allografts into Fascin1-null mice (Ma 2013). This suggests that targeting fascin would have limited side-effects in patients, as also noted by (Ma 2014). We now included this information and the pertaining References at the end of the discussion (lines 323-327):

“Fascin inhibitors have been developed that can be used in vitro, and more recently also in vivo (47,77–79), and inhibition of Fascin1 is generally compatible with adult tissue homeostasis (76,80,81). This suggests that Fascin inhibitors may represent in future an interesting approach to help disable iCCA malignancy.”

Minor points:

-In Figure 1F, please indicate the different doses of G2 on the graph itself to differentiate which one is which.

This has been done.

-In Fig2F, Fascin staining should be added. Showing only GFP creates a confusion that GFP tagged protein was expressed but overexpressed protein is not tagged (It is GFP-IRES-Fascin).

We are sorry for the misunderstanding. We had indicated in the methods the precise nature of the plasmid, i.e. Fascin-IRES-GFP (lines 381-382), so the GFP channel represents a lineage tracer for transfected cells. This set-up has the following advantages: the GFP is not fused to the protein of interest, so that it cannot potentially alter its function; the GFP is expressed from the most 3' open-reading frame of a bicistronic mRNA, so that if GFP is expressed, also the upstream protein is co-expressed in the same cell. We now changed the annotation of the figures (GFP+FSCN1) and figure legends accordingly, to avoid confusion.

-Graph representation in Sup Fig4 needs improvement. It is impossible to read the labels on both axes. Also, are these cell lines YAP-dependent? It might be good to see a control related to YAP here.

Thanks for pointing out this. We have corrected the labels (now in Fig. 4D). Moreover, in response to the Reviewer request, we performed EdU incorporation upon YAP/TAZ knockdown to show YAP-dependency in the same lines (Fig. 4E), and we additionally provide data of Fascin-dependent YAP localization on selected cell lines (Fig. 4F). We are sorry we could not provide the proliferation data with the same technique, but those data were originally obtained by our colleagues in Germany, but since they have experienced a complete lockdown in these months we had the cells shipped in Italy to perform the experiments. We moved this whole set of data in the main Fig. 4 as we believe this strengthens the evidence for a role of Fascin1 in human cholangiocarcinoma.

(data in the following page)

Legends: D. Real-time growth curve of four different human intrahepatic cholangiocarcinoma (iCCA) cell lines treated with different doses of G2 Fascin inhibitor, as measured by the xCELLigence Real Time Cell Analysis system and expressed as Normalized Cell Index. E. EdU incorporation assay of four different human iCCA cell lines transfected with control (siCo.) or validated YAP/TAZ siRNA (siY/T), 48 hours after transfection. $n=2$ (>200 cells per condition in total). F. Representative immunofluorescence for YAP/TAZ in KKKU-156 iCCA cells transfected with control (siCo.) or with Fascin1 siRNAs (siFSCN1 A and B). On the right: quantification of YAP/TAZ nuclear localization in KKKU-156 and RBE cells. $n=2$ (>50 cells per condition in total).

Reviewer #3 (Remarks to the Author):

The manuscript by Pocaterra et al., describes a novel function for Fascin1 in YAP mechano transduction. The authors start by characterizing the role of proteins such as the formins, Ena/VASP and Fascin in regulation of YAP. Significantly, the authors show that inhibition of Fascin by knockdown or small molecule leads to accumulation of YAP/TAZ in the cytoplasm and reduced transcriptional activity. In addition, they demonstrate that inhibition of Arp2/3 reduces YAP/TAZ nuclear localization, which phenocopies CAPZ depletion. Moreover, this appears to be mediated through Fascin1. The authors were able to corroborate some of their findings in vivo, by overexpressing Fascin1 in the liver and demonstrating this effect requires YAP/TAZ. In support of the function of Fascin1 as an activator of YAP/TAZ, the expression of Fascin1 along with activated AKT in the mouse liver leads to development of CCA. Using different genetic combinations to induce CCA, the authors observe Fascin1 upregulation in the myrAKT/NICD model. This is of interest since this combination relies on YAP for induction of CCA. Indeed, inhibition of Fascin1 in this model inhibited CCA development. Finally, the authors interrogated the expression of Fascin1 in liver tumors from patients and find increased levels in CCA and to a lesser extent in HCC.

This is an interesting report that is appropriately controlled. It provides interesting observations regarding the mechanisms by which mechanotransduction regulates signaling through YAP and provides a relevant in vivo context.

We thank the Reviewer for support. In response to the Reviewers, we carried out several experiments, provided better analyses, and quantified more precisely some phenotypes. In response to Reviewer #1, we now provide a better characterization of the lesions induced by the AKT+FSCN1 (new Fig. 3D-E), new data on the YAP/TAZ-dependency of human iCCA cell lines complementing previous data on fascin dependency (new Fig. 4D-E), and new data on the Fascin1-dependency of YAP/TAZ nuclear localization in human iCCA cell lines (new Fig. 4F). We think this helped make our story more convincing, and our claims more precise. We hope this Reviewer will continue supporting us, and will find our manuscript adequate for publication.

My suggestions are minor and listed below:

1. The effects of G2 treatment in vivo, as shown in Figure 3A, are subtle. One contributing factor could be the IP method of delivery. Since the authors have experience with HDTV injection, it might be worthwhile trying this as a method of delivery to enhance the impact of G2 treatment.

We used IP delivery method because this had been previously used in the relevant literature, and because changing it would have required dose-escalation experiments for which we have no current permissions. It is true that G2 treatment provides subtle effects, but please consider this compound was not optimized for in vivo studies. A better Fascin inhibitor was published late last year (Wang 2020), however we could not perform new in vivo experiments due to restrictions related to the pandemics. We are sorry for that.

2. While this might be beyond the scope of the manuscript, it would be interesting to assess the function of Fascin1 in the same cell type grown on substrates that have a stiffness gradient.

We haven't performed this experiment exactly, but we provide new evidence that Fascin expression in epithelial cells surrounded by wild-type cells in a confluent situation, is sufficient to induce YAP nuclear localization (Supplementary Fig. 2G). This builds on the notion that formation of confluent monolayers entails the reduction of cell-ECM adhesion area and the activation of what we called a

“mechanical checkpoint” similar to that occurring on a soft ECM, such that inactivation of CAPZ can rescue YAP activity in both conditions (Aragona Cell 2013; Benham-Pyle Science 2015). We think this is important to reinforce the CAPZ/Fascin antagonism idea, and to expand the relevance of our findings with an independent experimental set-up. Moreover, we also show in this experimental set-up that Fascin expression leads to increased proliferation in combination with oncogenic activation (Supplementary Fig. 2G). Finally, we now introduced in the discussion the idea that the effects of Fascin (either loss or gain) might be particularly relevant at intermediate levels of tissue stiffness, where YAP/TAZ activity is at an intermediate level (lines 284-292):

“Our data suggests that such competition may also occurs during ECM mechanotransduction, and becomes relevant when cells are in conditions of decreased ECM stiffness: in these conditions, not only RHO signalling is decreased, leading to decreased activity of ROCK/MLCK and of Formins (1), but at the same time CAPZ-dependent and Arp2/3-dependent actin outbalances actin bundles promoted by Ena/VASP, Fascin1 and potentially other F-actin bundling proteins. We speculate this might be particularly relevant at intermediate stiffness levels, ultimately tipping the balance in favor or against YAP/TAZ activity.”

Legend: Representative immunofluorescence images of MCF10A cells transiently transfected with the indicated combinations of plasmids encoding for GFP, FSCN1, H-RAS G12V, and myristoylated AKT were seeded together with non-transfected cells at high density. After 48 hours, cells were harvested for analysis of YAP/TAZ localization and EdU incorporation (quantifications on the right). Dotted white lines indicate transfected cells. YAP/TAZ and EdU levels were similar in GFP and in surrounding non-transfected cells. Data are mean and single points. n=2 experiments.

REVIEWERS' COMMENTS:

Reviewer #2 (Remarks to the Author):

The authors have done a good job of responding to my comments.

Reviewer #3 (Remarks to the Author):

The authors did an excellent job in responding to the review comments.